# Exploring temperature and humidity environment combined with air quality index, black carbon, the short-term effect of combined exposure on respiratory disease mortality in Southwest China

**Hengyu Su[1‡], Di Wu[1], Song Chen[2], Kaiyang Guo[3], Huifang Xie[1]***

**1** School of Public Health, Xinjiang Medical University, Urumqi, Xinjiang, China, **2** School of Health Management, Xinjiang Medical University, Urumqi, Xinjiang, China, **3** School of Nursing, Xinjiang Medical University, Urumqi, Xinjiang, China

‡ This is the independent first author.
* xhfworld@sina.com

## Abstract

This study investigates the correlation, impact, and hysteresis effect of joint exposure to the Temperature-Humidity Index (THI), Air Quality Index (AQI), and Black Carbon (BC) on respiratory disease mortality (RDM) in urban areas of the southwest basin of China, characterized by a subtropical monsoon climate. Dose-response analysis of THI, AQI, BC using a non-restrictive cubic spline model, a time series analysis was conducted to assess the relative risk (RR) of death from respiratory diseases using the distributed lag nonlinear model (DLNM) and the generalized additive model (GAM) based on the quasi-Poisson distribution. The RCS curve of THI exhibits a 'U' shape, with THI=67 representing the lowest point of mortality risk. The RCS curves for BC and AQI are linear and demonstrate a positive correlation with mortality outcomes. The peak mortality risk associated with the AQI typically occurs at Lag 2-3, with T3A3 (THI ≥ 75 and AQI ≥ P90) contributing to the highest excess mortality [excess increased risk rate (ER) = 0.55, 95% CI: 0.20, 0.81]. The peak risk of mortality associated with BC occurs at Lag0, with the highest excess mortality resulting from T3B3 (THI ≥ 75 and BC ≥ P90) combined events (ER=0.28, 95% CI: 0.10, 0.58). The cumulative relative risk (CRR) was highest in T3, with the peak CRR of 3.99 (95% CI: 1.26, 7.11) observed in definition T3A3. The relative risk of interaction (RERI) reveals varying degrees of positive additive interactions (RERI > 0) among AQI, BC, and THI.

## Introduction

In the context of global climate change, air pollution and significant alterations in climate are primary factors contributing to the morbidity and mortality associated with respiratory diseases [1]. Previous studies have demonstrated that the combined effects of climate change and air pollution on respiratory diseases are significant and non-linear [2].

**Data availability statement:** Data are not publicly available due to confidentiality agreements and government restrictions on the identifiable nature of the data, but are available upon reasonable request from Naifan Zhang (send requests to 1830487645@qq.com). All results generated throughout the study are included in the Supplementary Material, which is directly accessible.

**Funding:** The authors acknowledge that the research was supported by internal funds provided by School of Public Health, Xinjiang Medical University. These internal funds were used to cover the costs of conducting the study, including materials, equipment, and personnel expenses. No external funding was received, and there are no conflicts of interest arising from the use of these internal funds. The authors also acknowledge the support of the 14-th Five-Year Plan Distinctive Program of Public Health and Preventive Medicine in Higher Education Institutions of Xinjiang Uygur Autonomous Region. This fund is designated as the project support program fund for the school college, and our corresponding author is the recipient of this funding.

**Competing interests:** The authors have declared that no competing interests exist.

As a significant component of $PM_{2.5}$, Black Carbon poses a substantial threat to human health and is particularly associated with respiratory diseases. Long-term exposure to BC can lead to symptoms such as coughing, chest pain, drowsiness, and skin irritation [3]. Furthermore, it can result in a range of respiratory diseases, including lung damage [4]. The International Agency for Research on Cancer (IARC) has classified BC as a Category 2B carcinogen [5].

Previous studies have primarily concentrated on the health effects of temperature, including extreme climate events such as heat waves and cold waves, while treating meteorological indicators like humidity as confounding factors. Indeed, meteorological factors, such as humidity, have a modifying effect on the health impacts of temperature, as they influence human perception of temperature [6,7]. To the best of our knowledge, the Temperature-Humidity Index (THI) is utilized for public health monitoring in the United States and several European countries. In Japan, THI is employed to develop an early warning system for heat waves, which assesses and manages the effects of warm and humid weather, as well as urban heat island phenomena, on the health of residents. In China, THI is predominantly utilized in agriculture, animal husbandry, and tourism. However, there is a scarcity of studies and applications addressing the respiratory health of residents in conjunction with atmospheric pollutants in the field of public health. Most research regarding the effects of air pollution and climate change on respiratory diseases is primarily focused on eastern China, particularly in the Yangtze River Delta, the Pearl River Delta, and the southeastern coastal regions [8–10]. There are fewer studies conducted in southwest China, and no research has specifically defined the impact of joint exposure to Temperature and Humidity Indices, along with atmospheric pollutants, on respiratory disease mortality in unique geographical environments.

Consequently, it is essential to integrate the thresholds of local geographical and meteorological characteristics to establish a relevant definition of combined events. This study investigates the joint and interactive effects of temperature, humidity, and atmospheric pollutants on respiratory disease mortality in a region of southwest China. It aims to identify critical combinations of risk exposures under various definitions. Provide evidence to understand the health risks of mixed exposure to BC and AQI on respiratory diseases in warm and humid environments. This study employs the THI to integrate temperature and humidity, thereby providing a more comprehensive definition and modeling analysis of the environmental conditions. Additionally, it aims to establish a targeted health warning system to mitigate exposure risks. The definition and identification of exposure events are critical prerequisites for the development of an effective health early warning system.

## Materials and methods

### Research site

The study area is situated in the Sichuan Basin, China, specifically in the southwest region of the Chengdu Plain. It experiences a humid subtropical climate characterized by high temperatures and humidity throughout the year. Surrounded by mountains, this area faces significant pollution challenges, making it one of the most polluted regions in China. As of the end of 2023, the permanent population was recorded at 2.955 million, with men and women each representing 50.00% of the total. The demographic distribution shows that individuals aged 0-14 comprise 14.01% of the population, those aged 15-59 account for 61.26%, and the population aged 60 and above constitutes 24.73%, with individuals aged 65 and above making up 20.02%.

### Data collection

Considering the availability and timeliness of mortality, air pollution, and meteorological data, we obtained daily death records from January 2018 to December 2023 from the regional

Centers for Disease Control and Prevention system. This dataset includes the date of death, the ethnicity of the deceased, household registration details, the direct cause of death, and the underlying cause of death. Two experts from the Centers for Disease Control and Prevention screened the data according to the codes of the International Statistical Classification of Diseases (10th edition, ICD-10) pertaining to respiratory diseases (J00-J99), to ensure the accuracy and scientific integrity of the data.

**Meteorological data source.** The European Center for Medium-Range Weather Forecasts (ECMWF) (https://cds.climate.copernicus.eu/) provides data on daily average temperature, daily maximum temperature, daily minimum temperature, relative humidity, wind speed, and surface pressure. The THI serves as an indicator that integrates daily average temperature and relative humidity to assess the heat of the climate [11]. The Copernicus Atmosphere Monitoring Service (CAMS) provides daily data on various air pollutants, including $PM_{2.5}$, $PM_{10}$, $SO_2$, $CO$, $O_3$, $NO_2$, BC, and the AQI. The accuracy of ECMWF data can range from 70% to 90%, with information gathered through a global meteorological observation network that includes ground weather stations, ocean buoys, and meteorological satellites. The CAMS collects atmospheric composition data using various methods, including satellite remote sensing technology, ground monitoring stations, weather balloons, and meteorological models. Its high precision and accurate spatial matching contribute to the reliability of research findings. Both meteorological and air pollutant data were $1 \times 1km$ raster data, and patient addresses were API matched to calculate individual exposure for each patient.

**Data quality control.** We meticulously cleaned, screened, and organized the data in alignment with the objectives of the research. Initially, specific time periods and diseases were identified, and data were excluded for individuals who were not present in the area during their lifetime, as well as for those who resided in the area for less than five years prior to their death. We subsequently removed any missing values and cross-verified values that deviated by more than three standard deviations from the long-term mean. Finally, death statistics are aligned by date with air pollutant and meteorological data.

## Statistical analysis

**Definitions of exposure events.** The definition of exposure events is informed by the RCS exposure response curve, in conjunction with the 'GB3095-2012' standard established by the World Meteorological Organization (WMO), the National Meteorological Administration, and the China Meteorological Administration for heat wave definitions, as well as previous research. We categorize the Temperature-Humidity Index (THI) as follows: T1 (THI < 63), T2 ($63 \leq$ THI < 75), and T3 (THI $\geq$ 75). Furthermore, by integrating the BC and AQI percentiles from this study and referencing the Technical Specifications for Ambient Air Quality Assessment (HJ6632013), the Guidelines for the Preparation of Urban Heavy Air Pollution Emergency Plans, and the Sichuan Province Heavy Pollution Weather Emergency Plan (2018 Revision) - Quantitative Indicators, we redefine the BC and AQI categories as B1 ($P \geq 50$), B2 ($P \geq 75$), B3 ($P \geq 90$), A1 ($P \geq 50$), A2 ($P \geq 75$), and A3 ($P \geq 90$). Notably, A1 encompasses A2 and A3, while B1 includes B2 and B3. This study utilizes the RCS intersection value as a node, treating the combined meteorological event variable as a continuous variable, thereby fixing it within the ranges above or below different percentiles. This approach aims to investigate the relationship between THI and extreme AQI and BC, as well as to explore the impact of exposure to combined events with varying threshold ranges on respiratory mortality, ultimately refining the thresholds for early warning models.

The comprehensive event definition comprises three THI, along with three definitions for two pollutants, resulting in a total of 18 distinct definitions. The combined event, denoted as

XiYj, is defined by the individual definitions of Xi and Yj, indicating that both events occur simultaneously.

**Outult of exposure events.** The combined meteorological event variables were the continuous variables. The THI, AQI, BC, and mortality data did not conform to a normal distribution upon testing. Consequently, the statistical package of R version 4.3.0 was utilized to perform Spearman correlation analysis to assess the relationships between each variable. This analysis served as a foundation for the subsequent establishment of GAM and DLNM. This study employs a time series analysis that integrates a GAM and a DLNM utilizing a quasi-Poisson distribution. Initially, we employ the GAM to analyze nonlinear relationships within time series data, effectively fitting potential patterns and trends using various smoothing functions. Subsequently, we utilize the DLNM to dynamically capture the temporal and dose-response effects, addressing the intricate relationships between environmental factors and health outcomes. This approach culminates in the establishment of an exposure-lag-response model. When used in conjunction, the two can fully leverage their complementary characteristics, thereby enhancing the novelty, depth, and breadth of the analysis. The objective is to estimate the exposure-lag-response relationship among THI, AQI, BC, and Respiratory Disease Mortality (RDM). The analysis includes plotting specific incremental effects and the CRR associated with these variables. Additionally, the hysteresis response curve is examined to illustrate the effects. The results from the additive interaction model are used to assess the existence of interactions.

**Model building.** The calculation formula for THI was established from previous studies [12,13]:

$$THI = \left(1.8 * T_{avg} + 32\right) - \left[\left(0.55 - 0.0055 * RH_{avg}\right) * \left(1.8 * T_{avg} - 26\right)\right] \tag{1}$$

$T_{avg}$ is the daily mean temperature (°C) and $RH_{avg}$ is the daily average relative humidity (%).

Utilizing the 'rms' package, we will establish restricted cubic splines to investigate the dose-response relationship between the THI, AQI, BC, and mortality outcomes. This analysis aims to identify the threshold cut-off points for each factor concerning respiratory disease mortality and to define exposure events. The fundamental expression of the model is presented as follows [14]:

$$Y = \beta 0 + \beta 1 X 1 + \beta 2 X 2 + \ldots + \beta n X n + \varepsilon$$

Y stands for the dependent variable (response variable). β0 represents the intercept, which represents the predicted value of the dependent variable Y when all independent variables X are zero. β1, β2,..., βk represent the slope coefficients, which represent the expected change in the dependent variable Y when an independent variable Xi is increased by one unit. Each coefficient represents the degree to which the independent variable affects the dependent variable. X, X1, X2,..., Xk represent the independent variable (explanatory variable) and are used to predict the dependent variable Y. ε represents the error term, which represents the difference between the predicted value of the model and the actual observed value.

We incorporated the cross basis function of the combined event into the model, selecting the longest lag period (lag) of 7 days based on prior studies [8,9], as well as the AIC, to effectively capture the overall impact of the combined event while controlling for temporal trends and other confounding factors. The fundamental algebraic expression of this model is provided as follows [8]:

$$Y_t \sim Poisson(\mu_t)$$

$$\text{Log}\left(\mu_t\right) = \alpha + \beta T_t, l\left(Q\right) + ns\left(\text{time, df}\right) + ns\left(\text{meteorology, df}\right) + y\text{DOWt} + \lambda\text{Holidayt} \quad (2)$$

In the formula, Log represents the connection function; $y_t$ denotes the number of RDM individuals on day t; α is the intercept; β, y, and λ are the corresponding regression coefficients; Q signifies the combined event; $l$ indicates the longest lag number of days; $T_t, l$ refers to the combined event cross basis matrix; ns is the cubic spline function employed to account for unmeasured confounding factors related to time, seasonality, and long-term trends; 'time' is the time variable, with df set to 7 per year, which is utilized to smooth out seasonality and cycles of sexual influence; 'meteorology' encompasses other meteorological factors, with df set to 3; DOWt represents the day of the week for the t-th day, primarily used to control for the day-of-the-week effect; and Holidayt indicates whether the t-th day is a statutory holiday or falls within winter and summer vacations.

In this study, THI, AQI, and BC levels were increased by one interquartile range (IQR) to investigate the relationship between changes in their concentrations and mortality outcomes. The results are reported using the relative risk of death (RR, 95% CI) and the excess increased risk rate (ER, 95% CI) [8,9,11].

$$\text{IQR} = Q3 - Q1$$

$$\text{RR} = \exp\left(\text{IQR} * \beta\right)$$

$$\text{ER} = \left(\text{RR} - 1\right) * 100\%$$

We utilize the relative risk due to interaction (RERI) to assess the comprehensive impact of the THI, AQI, and BC. A RERI of 0 signifies no additional risk, while a RERI greater than 0 indicates that the combined effects of the THI, along with gaseous pollutants, on health are greater than the individual effects of each factor. Conversely, a RERI less than 0 suggests that the joint effects are less than additive.

$$\text{RERI} = RR_{11} - RR_{01} - RR_{10} + 1$$

Among these, $RR_{11}$ represents the relative risk of the combined event, $RR_{01}$ denotes the relative risk associated with THI, and $RR_{10}$ indicates the relative risk related to the AQI or BC event.

The cumulative effect of combined events on mortality over a lag period of 0 to 7 days was defined as the cumulative relative risk (CRR, 95% CI). This study estimates the single-day lagged effects from lag 0 to lag 7, where lag 0 and lag 1 correspond to the lagged effects of exposure factors on respiratory disease deaths on the first and second days, respectively. Additionally, we evaluated the cumulative lagged effect from lag 01 to lag 07, which reflects the cumulative lagged impact of exposure factors on disease mortality from day 2 to day 1.

**Sensitivity analysis.** To assess the robustness of our modeling strategy, we conducted a sensitivity analysis by varying the degrees of freedom (df) for factors 3, 5, and 6 in several key models. All analyses were performed using R software version 4.3.0, employing the 'splines' and 'lme4' packages to carry out stratified analyses by gender, age, and other variables. A p-value of less than 0.05 was considered statistically significant.

## Result

### Descriptive analysis

Table 1 presents a summary of respiratory disease mortality, the AQI, the THI, and black carbon levels in the region from January 1, 2018, to December 31, 2023. During this period, a

**Table 1. Indicators of the number of deaths, air pollutants and meteorological factors in a city, 2018-2023.**

| Variables | $\overline{x} \pm s$ | Min | P25 | P50 | P75 | Max |
|---|---|---|---|---|---|---|
| All-death | 6.73 ± 5.32 | 0.00 | 3.00 | 6.00 | 10.00 | 51.00 |
| $PM_{2.5}$ (μg/m³) | 109.50 ± 30.18 | 44.13 | 85.73 | 103.95 | 128.32 | 337.58 |
| $PM_{10}$ (μg/m³) | 152.77 ± 46.51 | 61.71 | 120.28 | 145.18 | 179.33 | 445.83 |
| $SO_2$ (μg/m³) | 33.73 ± 5.23 | 17.41 | 29.89 | 33.59 | 37.48 | 51.05 |
| $NO_2$ (μg/m³) | 41.54 ± 9.01 | 18.39 | 35.46 | 40.13 | 46.22 | 107.33 |
| $O_{38h}$ (μg/m³) | 48.49 ± 21.10 | 4.63 | 31.93 | 45.08 | 64.41 | 121.59 |
| CO (mg/m³) | 1.03 ± 0.34 | 0.32 | 0.78 | 0.95 | 1.23 | 2.91 |
| BC(μg/m³) | 1.64 ± 0.58 | 0.55 | 1.19 | 1.53 | 1.98 | 5.70 |
| Average daily temperature (°C) | 18.26 ± 7.46 | 1.74 | 11.71 | 18.58 | 24.67 | 36.12 |
| Ground pressure (hPa) | 960.42 ± 9.73 | 933.96 | 953.41 | 960.66 | 968.41 | 985.22 |
| Wind speed (m/s) | 1.60 ± 0.66 | 0.02 | 1.23 | 1.54 | 1.95 | 4.83 |
| Relative humidity (%) | 75.81 ± 10.97 | 22.24 | 69.33 | 77.48 | 83.79 | 96.46 |
| THI | 63.70 ± 11.47 | 37.12 | 53.66 | 64.24 | 73.63 | 84.91 |
| AQI | 76.50 ± 32.24 | 30.00 | 62.00 | 80.00 | 106.00 | 232.00 |

total of 13,847 individuals succumbed to respiratory diseases, with daily mortality rates fluctuating between 0 and 51. The average THI value recorded in the area is 63.70, with a range of 53.66 to 73.63. The average AQI value is 76.50, spanning from 62.00 to 106.00, while the average BC concentration is 1.64, with values ranging from 1.19 to 1.98.

## Restrictive cubic spline results

Fig 1 presents the results of the restricted cubic spline analysis for the THI, AQI, and BC in relation to respiratory disease mortality. All three variables exhibit an exposure-response relationship with the mortality outcome. Notably, the exposure-response curve for THI is characterized by a nonlinear, U-shaped pattern, with a minimum risk of death observed at a THI value of 67. In contrast, the relationships between AQI, BC, and the mortality outcome are linear.

## Definition and number of indicators

Based on the results of the restricted cubic spline presented in Fig 2, Table 2 provides the relevant definitions and the number of independent exposure events. Notably, T1 exhibits the highest number of events as defined by the THI.

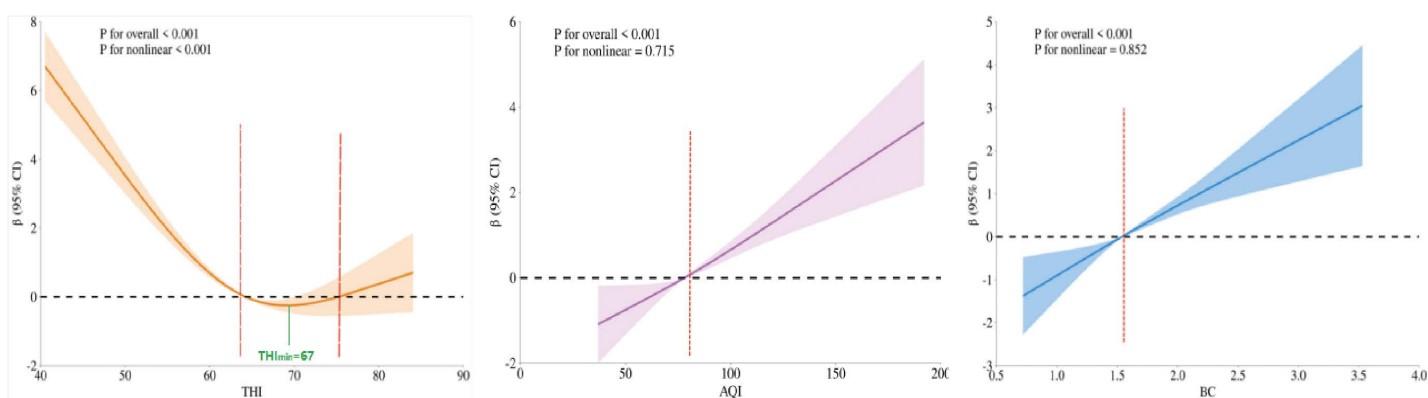

**Fig 1. Results of restricted cubic spline analysis of THI, AQI, and BC in this region.**

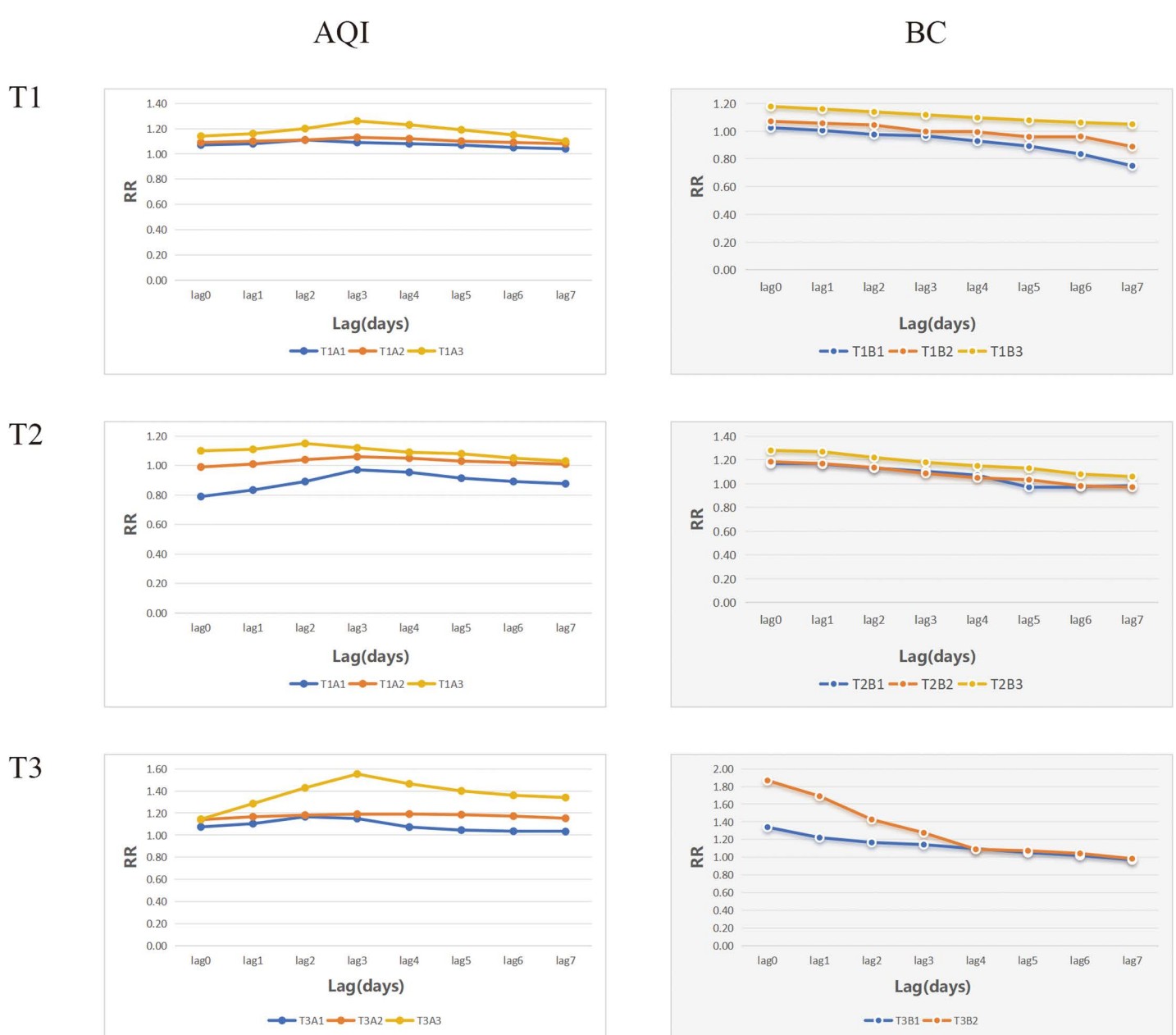

**Fig 2. Exposure lag effect curves for the combined events of AQI, BC and THI in this region.** Note: Combined event definitions refer to the independent definitions presented in Table 2; for instance, T1A3 denotes the combined event defined by T1 and A3.

## Combination events

Table 3 presents the number of merged events, revealing that the maximum count under the T1 definition is 739, categorized as T1B1. In contrast, the highest number of merge events in a T2 scenario is 311, designated as T2A1. For the T3 definition, the largest number of combined events is found in T3A1, totaling 302. To ensure an adequate sample size, events with durations exceeding 10 days were filtered to further assess health effects.

**Table 2. Index definition range of THI, AQI, and BC in this region.**

| Event | Index | Definition | Number |
|---|---|---|---|
| THI | T1 | THI < 63 | 973 |
| | T2 | 63 ≤ THI < 75 | 723 |
| | T3 | THI ≥ 75 | 498 |
| AQI | A1 | AQI ≥ 80(P50), with ≥ 1-days duration | 1115 |
| | A2 | AQI ≥ 106(P75), with ≥ 1-days duration | 551 |
| | A3 | AQI ≥ 130(P90), with ≥ 1-days duration | 232 |
| BC | B1 | BC ≥ 1.53(P50) | 1100 |
| | B2 | BC ≥ 1.98(P75) | 555 |
| | B3 | BC ≥ 2.45(P90) | 220 |

**Table 3. Days of combined events in the region.**

| | A1 | A2 | A3 | B1 | B2 | B3 |
|---|---|---|---|---|---|---|
| T1 | 502 | 277 | 108 | 739 | 396 | 192 |
| T2 | 311 | 107 | 50 | 293 | 111 | 27 |
| T3 | 302 | 167 | 74 | 68 | 48 | 1 |

**Table 4. Results of spearman correlation among various factors in this region.**

| | $PM_{2.5}$ | $PM_{10}$ | $O_3$ | CO | $SO_2$ | $NO_2$ | Wind | AP | AQI | BC | THI |
|---|---|---|---|---|---|---|---|---|---|---|---|
| $PM_{2.5}$ | 1 | | | | | | | | | | |
| $PM_{10}$ | 0.99** | 1 | | | | | | | | | |
| $O_3$ | -0.33** | -0.32** | 1 | | | | | | | | |
| CO | 0.71** | 0.73** | -0.27** | 1 | | | | | | | |
| $SO_2$ | 0.63** | 0.64** | -0.21** | 0.76** | 1 | | | | | | |
| $NO_2$ | 0.59** | 0.60** | -0.35** | 0.75** | 0.56** | 1 | | | | | |
| Wind | -0.07** | -0.08** | -0.08** | -0.18** | -0.22** | -0.12** | 1 | | | | |
| AP | 0.33** | 0.33** | -0.63** | 0.26** | 0.15** | 0.20** | 0.25** | 1 | | | |
| AQI | 0.28** | 0.28** | 0.21** | 0.39** | 0.27** | 0.20** | -0.16** | -0.18** | 1 | | |
| BC | 0.87** | 0.88** | -0.31** | 0.77** | 0.56** | 0.70** | -0.05* | 0.34** | 0.25** | 1 | |
| THI | -0.50** | -0.49** | 0.75** | -0.46** | -0.30** | -0.38** | -0.04 | -0.75** | 0.02 | -0.54** | 1 |

*Note*: AP: air pressure.

**$P < 0.05$.

## Spearman correlation analysis

Table 4 presents the results of the Spearman correlation analysis. Among the variables analyzed, $PM_{2.5}$ and $PM_{10}$ exhibit the highest correlation, with a coefficient of r = 0.99. With the exception of the correlations between THI and wind speed, as well as THI and AQI, all other pairs demonstrate varying degrees of correlation that are statistically significant.

## Impact of the combined exposure events

Fig 2 illustrates the lag structure associated with the combined events of respiratory death. The RRs along with their 95% confidence intervals are detailed in S1–S6 Tables. It is important to highlight that the effects of various scenarios and causes exhibit significant variability.

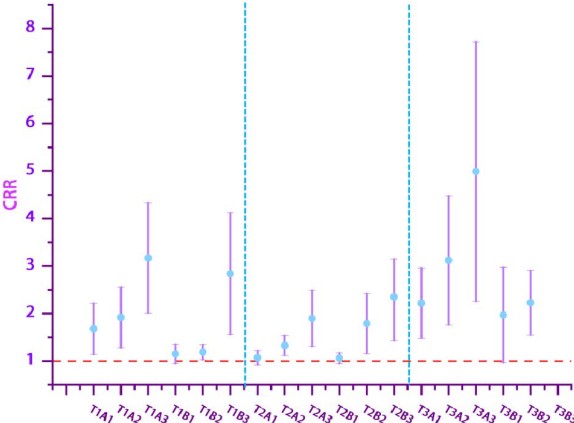

**Fig 3. Cumulative relative risk of AQI, BC and THI combination events in this region.** Note: Combined event definitions refer to the independent definitions presented in Table 2; for instance, T1A3 denotes the combined event defined by T1 and A3.

In the analysis of respiratory disease mortality outcomes (Fig 2), the peak of the hysteresis curve for the combined events of T1, T2, T3, and AQI typically occurs during lag 2-3, exhibiting an overall "inverted V" shape. Among the various T1 definitions, T1A3 resulted in the highest excess mortality during lag 3, with an RR of 1.26 (95% CI: 1.11, 1.41). For T2, T2A3 recorded the highest RR during lag 2, at 1.15 (95% CI: 1.04, 1.26). In the T3 definition, T3A3 showed the highest RR during lag 3, measuring 1.55 (95% CI: 1.20, 1.81).

Under different definitions, the combined event of Temperature-Humidity Index (THI) and black carbon reveals a hysteresis curve that exhibits a gradual downward trend, with peaks occurring during the lag0 period. Among the three definitions of THI, T1B3, T2B3, and T3B2 demonstrate the highest excess mortality risks, recorded at 1.18 (95% CI: 1.04, 1.33), 1.28 (95% CI: 1.12, 1.45), and 1.87 (95% CI: 1.21, 2.53), respectively.

When exposed to the same increment, the relative risk (RR) of the combined event defined by T3, AQI, and BC is generally higher than that of the combination of T1 and T2.

## CRR result

Fig 3 illustrates the CRR for lags of 0–7 days. Notably, the CRR for the T3 segment is greater than that for both T1 and T2. The CRR for the AQI varies from T2A1 (1.07; 95% CI: 0.92, 1.22) to T3A3 (4.99; 95% CI: 2.26, 7.72). The CRR for BC ranges from T2B1 (1.06; 95% CI: 0.95, 1.17) to T1B3 (2.84; 95% CI: 1.56, 4.12).

## RERI result

Synergistic effects of exposure events on respiratory disease mortality can be observed in combination events characterized by extremely high concentrations of pollutants and THI. Six definitions (T1A3, T2A3, T3A3, T1B3, T2B3, T3B2) were identified as having a synergistic effect on respiratory mortality, particularly in T3 (Table 5).

## Sensitivity analysis

The AIC (or BIC) values presented in S7 Table indicate that the number of nodes for THI, BC, and AQI are 3, 3, and 4, respectively. Furthermore, the sensitivity analysis results for the remaining definitions demonstrate robustness, as shown in S8 and S9 Tables.

Table 5. RERI calculated for combined events in this region.

| RRs | RERI(95%CI) | RRs | RERI(95%CI) | RRs | RERI(95%CI) |
|---|---|---|---|---|---|
| T1A1 | 0.05 (-0.01, 0.11) | T2A1 | -0.08 (-0.10, -0.06) | T3A1 | 0.01 (-0.01, 0.04) |
| T1A2 | **0.03 (0.02, 0.04)** | T2A2 | 0.01 (-0.02, 0.04) | T3A2 | **0.05 (0.02, 0.08)** |
| T1A3 | **0.04 (0.01, 0.07)** | T2A3 | **0.02 (0.00, 0.04)** | T3A3 | **0.06 (0.04, 0.08)** |
| T1B1 | -0.01 (-0.04, 0.02) | T2B1 | 0.02 (-0.02, 0.06) | T3B1 | 0.01 (-0.02, 0.04) |
| T1B2 | **0.02 (0.01, 0.03)** | T2B2 | 0.01 (-0.03, 0.05) | T3B2 | **0.08 (0.01, 0.15)** |
| T1B3 | **0.03 (0.01, 0.05)** | T2B3 | **0.04 (0.02, 0.06)** | T3B3 | — |

Note: Bold words are definitions that highlight synergistic effects, but do not determine if they are statistically significant.

## Discussion

To investigate the effects of combined exposure to THI, AQI, and BC on mortality from respiratory diseases, we systematically analyzed the impact of these combined exposures under 18 distinct definitions. To the best of our knowledge, this study is among the few that specifically defines a combined event involving the THI, AQI, and BC levels in China. Our findings indicate that the adverse health effects of combined events differ depending on the definitions used. The lagged structure of integrated events varies across definitional strata. Therefore, determining the appropriate definition of exposure events is crucial for health risk assessment and the development of early warning systems.

Currently, China employs a unified set of grading standards for the THI and AQI; however, these standards have not been utilized to establish a high-temperature early warning system, making it challenging to adapt to various climate zones. Consequently, we conducted a dose-response analysis of THI, AQI, and BC using a RCS model tailored to the specific local environment. In this analysis, we defined thresholds T1, T2, T3, A1, A2, A3, B1, B2, and B3. Similarly, in Shanghai, China, Liang employed the wet bulb globe temperature (WBGT) index to establish the minimum incidence threshold point, based on the exposure-response relationship [15]. It is important to note that the definition of the THI threshold in this study diverges from the traditional grading definition [11,16]. This may result from the uncertainty surrounding recent climate change and the specificities of different regions. The average concentration and variation range of BC observed in this study were lower than those reported in previous studies [17,18]. This discrepancy may be attributed to China's vigorous efforts to control air pollution in recent years. The results of this study indicate that black carbon exhibits significant harmful effects under the definitions of T1, T2, and T3. A study conducted in Beijing [18] also found that the interaction between temperature and BC increased the excess number of deaths from respiratory diseases by 1.80%. To our knowledge, there are limited articles that define subgroups and analyze interactions with THI concerning BC thresholds. Consequently, it is essential to establish an early warning system for THI and black carbon risk, both in this region and across China.

In this study, the relationship between the RCS curve of THI and respiratory disease mortality outcomes was found to be nonlinear, exhibiting a 'U' shape. Specifically, THI = 67 represented the lowest point of mortality risk. This finding aligns with the results of Zhang [11], where a THI of approximately 55 to 70 demonstrated a negative correlation with the risk of death from respiratory diseases within the population. This may be attributed to the existence of optimal intervals among temperature, humidity, and the resulting outcome; thus, there may also be an optimal region for the combined effects of these two factors. When the THI index falls below or exceeds specific thresholds (T1, T3), a greater change in the index correlates with an increased risk of death. Although the biological mechanisms underlying

thermohumidity events remain unclear, one possible explanation is that during low THI events, reduced humidity impairs human respiratory mucosal function and ciliary motility. This impairment facilitates the adherence of bacteria to the mucosa, thereby increasing the risk of infection [19]. An experiment demonstrates that mice exposed to low humidity exhibit impaired mucociliary clearance, diminished innate antiviral defenses, and compromised tissue repair, resulting in increased susceptibility to inflammasome caspase-mediated diseases [20]. During high THI events, elevated temperatures combined with high humidity can hinder the body's ability to dissipate heat. This may result in heat-related illnesses such as heat stroke and heat cramps, while also increasing the strain on the human respiratory system [21,22]. Similarly, Andrews [23] defined various heat exposure indices and found that a higher heat exposure index is associated with a greater risk to human survival. Liang also found that, compared with high temperature conditions, hot and humid weather in Shanghai was significantly associated with a higher risk of outpatient visits and an increased risk of all-cause morbidity [15]. However, these findings are inconsistent with those of Fallah Ghalhari G, who identified an inverse correlation between the heat stress excitation threshold and mortality [24]. This discrepancy may be attributed to the lower humidity levels in the Middle East, as well as variations in socioeconomic factors among the study samples. It is anticipated that excessive heat stress will become prevalent in low- and middle-income countries located in tropical or subtropical regions. There is no nonlinear relationship between the exposure-response curves of AQI, BC, and respiratory disease mortality outcomes, which aligns with the findings of most studies [3–5,17]. In the United States, the AQI is commonly employed to convey daily air quality information to the public. Cromar [25] discovered a positive association between AQI and respiratory disease incidence among adults in California. Some studies [26] have indicated that the estimated health impact of a 1 μg/m³ increase in black carbon exposure, as observed in time series studies, is greater than that of $PM_{10}$ or $PM_{2.5}$. When expressed as black carbon, the estimated increase in life expectancy resulting from reductions in black carbon concentrations is 4 to 9 times greater than that associated with equivalent changes in $PM_{2.5}$ mass. Animal experiments [27] have revealed that lung macrophages containing black carbon particles exhibit selective mitochondrial structural damage and a reduction in oxidative respiration. BC exposure metabolically rewires lung macrophages to promote immunosuppression and accelerates the development of lung cancer. Gao [28] also found that BC particles induce lung disease, characterized by an increased expression of factors related to inflammation, necrosis, and fibrosis. The detrimental effects of BC on the lungs involve the targeting of lysosomes. The research results indicate that the integration of the THI index and black carbon is highly significant for developing a risk assessment system for the heat wave early warning system in our country. It is crucial to delineate and define thresholds based on the exposure-response relationship specific to each region, thereby ensuring adaptability to local conditions. In unique environments, risk warnings should be implemented to effectively safeguard the lives and property of local residents.

In this study, combined events present varying degrees of health risks. Notably, the mortality risk associated with the combined event of THI and AQI typically peaks with a delay of 2 to 3 days. This finding aligns with the research conducted by Surit P [29], who identified a correlation between the AQI and hospital visits for pneumonia occurring the day following exposure (RR = 1.024, 95% CI: 1.003-1.046). Furthermore, the AQI is linked to an increase in ICU admissions and mortality rates associated with respiratory illnesses. This study found that, with the same increase in exposure, the mortality risk and cumulative mortality risk of respiratory diseases in the population caused by combined T3 and AQI events appear to be more severe. Luo [30] found that as the AQI index increases, the severity of air pollution, as classified by the AAQI and HAQI indices, also intensifies, leading to

a higher mortality rate from respiratory diseases. This study demonstrates a timely impact on the risk of death associated with the combined events of THI and BC. Under various definitions, higher concentrations of black carbon correlate with an increased cumulative risk of mortality. The cumulative mortality risk associated with the combined effects of T2 and black carbon is lower than that resulting from the combined exposure defined under T3. This discrepancy may be attributed to the influence of humidity, which alters the properties and pathogenic mechanisms of black carbon. Fang [31] found that, compared to weather characterized by normal temperature and high humidity, hot and dry weather poses a greater risk than hot and humid weather. This finding further elucidates the potential modifying effect of humidity. As humidity continues to rise, the risk of illness or death resulting from the combined effects of temperature and humidity also increases. This may be attributed to high humidity, which reduces the efficiency of heat dissipation through human pores, leading to decreased sweating and other related symptoms. Consequently, this increases the likelihood of heatstroke and other heat-related illnesses. This indirectly highlights that the overall mortality risk associated with combined exposure to high THI, AQI, and BC remains relatively high.

Studies [32] indicate that combined events have more severe health impacts than individual air pollution events. Furthermore, the results of this study demonstrate that simultaneous exposure to extreme THI and high pollutant events has a more significant impact on human health, aligning with the findings of previous research on combined events. In the definitions of T1, T2, and T3, the CRR values ranging from 0 to 7 for the combined events T1A2, T1A3, T2A2, T2A3, T3A2, and T3A3 in this region are consistently higher than those for T1A1, T2A1, and T3A1. This pattern is also observed in the combined events related to black carbon. The combined events of T3 and AQI demonstrated additive interactions, with the T3A3 combined event exhibiting the highest Relative Excess Risk due to Interaction (RERI). Additionally, the RERI for T2B3 was found to be the highest in the combined event associated with black carbon. The results indicate that, within the context of climate change, the impacts of pollutants and meteorological factors on human health are interdependent rather than isolated. Their combined exposure produces an additive effect on the human body, warranting significant attention. Consequently, in addressing air pollution control and public health prevention, it is imperative to focus on their comprehensive effects, as highlighted by various data analyses.

A stricter definition increases the relative risk; however, its protective capability remains limited. Conversely, a looser definition can encompass a greater number of risk exposure events, thereby offering broader protection, albeit with a comparatively lower relative risk [33]. However, mitigating risk events may inadvertently lead to an increase in warnings, while also escalating associated costs [34]. To balance the costs associated with the construction of early warning systems and the anticipated health benefits, further research is essential to explore appropriate definitions. THI is a dynamic metric influenced by temperature and humidity levels. Both BC and the AQI serve as monitoring results. Consequently, THI, BC, and AQI do not require seasonal division; rather, they should be defined solely by their respective numerical ranges. Thus, the findings of this study carry practical implications. Due to the significant lag effect associated with pollutants, there is also a corresponding lag effect on combined events, which peaks within the 0 to 3 lag days. This indicates a short-term effect; however, a strict definition of pollution appears to have an insignificant impact on health outcomes. Our findings indicate that when comparing individual pollutant exposures across various definitions of the THI, the results consistently demonstrate that the initial peak of lagged health effects does not occur earlier or later. This suggests a joint effect of AQI, BC, and THI events. The stability of these outcomes is beneficial for the issuance of early warning

announcements, the establishment of early warning systems, and the development and implementation of related policies.

The definition of the hygrothermal effect index differs from that of air pollution's impact on health outcomes [10], which may partially account for the relatively low health effects associated with T2. In light of the potential exposure lag response curve, cumulative relative risk results, and the effectiveness of risk early warning systems, we have thoroughly considered and discussed the establishment of an early warning system. Firstly, we recommend that the local climate environment, as well as economic and social conditions, be comprehensively evaluated. The thresholds for various influencing factors can vary significantly depending on the climate zone, and economic conditions may also influence the quality and standards of the early warning system. Secondly, given the hysteresis associated with the impacts, we propose extending the setting time for the early warning system and issuing alerts within one week following the occurrence of comprehensive exposure events to enhance population protection and prevention efforts. Furthermore, for the specific development and dissemination of the early warning system, we suggest collaboration with relevant agencies, such as the local CDC. This partnership should leverage the expertise of the CDC's infectious disease and environmental institutes, ensuring mutual support in providing technical assistance. Subsequently, relevant outreach departments should promote the system through various media channels to ensure that the majority of residents understand, recognize, and accept this system, thereby enhancing its practical implementation. Lastly, we recommend categorizing heat wave warning levels for exposure events into three tiers: low (T2), medium (T1), and high (T3). Given the heightened risks associated with combined events, we further propose dividing pollutant warnings into three categories based on the definitions of AQI and BC concentrations: '+Low', '+Medium', and '+High'. The THI's early warning system can be integrated with the existing heat wave early warning system. The simultaneous activation of similar components in both systems can reduce the costs associated with issuing early warnings. Moreover, the THI early warning system exhibits greater sensitivity in detecting changes in the external dynamic environment and takes into account the specific characteristics of the local environment. This provides relevant health departments with more scientific and accurate foundational information for effectively addressing heat and humidity incidents.

The calculation of the AQI primarily relies on monitoring the concentrations of various air pollutants. These pollutants typically include $PM_{2.5}$, $PM_{10}$, $O_3$, $SO_2$, and CO. BC is a component of $PM_{2.5}$; therefore, fluctuations in BC concentration are largely dependent on changes in $PM_{2.5}$ concentration. However, due to the rapid changes in climate, alterations in its material properties, and shifts in its pathogenic mechanisms, an increasing number of studies are now beginning to investigate its effects on the human body separately [35]. This is one of the reasons why BC is examined independently in this study. As a combination of exposure events, BC can be studied in conjunction with other pollutants such as $SO_2$, $NO_2$, and $O_3$ to investigate the effects of their combined exposure on the population. The environment we inhabit is not made up of a single substance or event. The occurrence of extreme temperatures, sunshine, precipitation, cyclones, and other weather phenomena frequently alters our lifestyle and surroundings. Relevant studies have demonstrated that short-term exposure to elevated levels of high temperatures, $NO_2$, $O_3$, and CO is significantly associated with an increased risk of respiratory mortality. Furthermore, men and the elderly are particularly susceptible to these environmental factors [36]. Du found that climate factors and air pollution have lagged and nonlinear effects on the epidemic of hand, foot, and mouth disease. Furthermore, the influence of climate factors on health may vary based on the AQI [37]. There is a connection between various weather types and air pollutants. For instance, coastal cities are more vulnerable to different patterns of atmospheric

circulation and types of cyclones [38]. In the context of climate change, it is essential to focus on the bidirectional interaction between meteorological conditions and air pollutants, as well as the effects of their combined exposure on the population. This emphasis underscores the significance of our research.

Our study presents several notable strengths. First, we investigated the impact of combined events using various definitions of THI, AQI, and BC, which offers robust epidemiological evidence regarding their influence on respiratory disease mortality in China. Second, we established a total of 18 distinct definitions. To our knowledge, the variability in the definition of combined events is greater than that observed in other relevant independent exposure studies. The construction of these definitions considers the full impact of exposure lag effects, and employing unrestricted cubic splines to delineate the ranges enhances both the scientific rigor and precision of our findings. Third, we have systematically proposed a comprehensive scientific basis and construction methodology for an early warning system, which provides valuable insights and frameworks for developing early warning systems in other regions, as well as across China as a whole.

This study has several potential limitations that warrant consideration. First, the research was conducted in a single study area and over a short time scale, which may introduce variability in the results. Additionally, this study primarily concentrated on short-term effects and did not fully investigate the long-term outcomes related to combined exposure events and respiratory deaths. This limitation may affect the comprehensiveness of the findings. To enhance and broaden the findings of this study, it is essential to incorporate additional research conducted in diverse climate zones, additional relevant studies are necessary to enhance the reliability of these findings. Second, we were unable to perform a more detailed analysis of the sensitive population due to a lack of information regarding their socioeconomic characteristics. Third, exposure data were not through individual monitoring, which may introduce biases in health risk assessments.

In summary, this study presents evidence of a correlation between an increased risk of mortality from respiratory diseases and combined environmental events. Raising awareness of the interplay between AQI, BC, and THI is crucial, as it can facilitate the management of associated health risks. Moreover, the integration of multiple definitions significantly improves the model's capacity to identify exposure events and can be adapted for combined exposure health risk assessments in other research domains.

## Supporting information

**S1 File**. **Supplementary Tables. Table S1.** Relative risk of combined events T1A1 T1A2 T1A3. **Table S2.** Relative risk of combined events T2A1 T2A2 T2A3. **Table S3.** Relative risk of combined events T3A1 T3A2 T3A3. **Table S4.** Relative risk of combined events T1B1 T1B2 T1B3. **Table S5.** Relative risk of combined events T2B1 T2B2 T2B3. **Table S6.** Relative risk of combined events T3B1 T3B2 T3B3. **Table S7.** The results of AIC(BIC) of combined events definition. **Table S8.** Sensitivity analysis for cumulative effect estimates of combined event (T3A1,T3B1) on respiratory mortality at lag 0 and lag 0-7 days. **Table S9.** Sensitivity analysis for cumulative effect estimates of combined event (T3A2,T3B2) on respiratory mortality at lag 0 and lag 0-7 days.
(DOCX)

## Author contributions

**Conceptualization:** Di Wu.

**Formal analysis:** Hengyu Su.

**Methodology:** Hengyu Su, Song Chen.

**Software:** Hengyu Su, Song Chen, Kaiyang Guo.

**Supervision:** Huifang Xie.

**Writing – original draft:** Hengyu Su.

**Writing – review & editing:** Hengyu Su.

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
