## [Decision Letter · Decision Letter 0]

17 Dec 2024

PONE-D-24-49828Exploring Health Effects under Specific Causes of Respiratory Mortality Based on Different Definitions of Air Quality Index,Black carbon,Temperature and Humidity environment Combined Exposure in Southwest,ChinaPLOS ONE

Dear Dr. Xie,

Thank you for submitting your manuscript to PLOS ONE. After careful consideration, we feel that it has merit but does not fully meet PLOS ONE’s publication criteria as it currently stands. Therefore, we invite you to submit a revised version of the manuscript that addresses the points raised during the review process.

**ACADEMIC EDITOR: ** The major negative aspects of the paper include the lack of novelty in methodology, as the study relies heavily on existing models like DLNM and GAM without substantial innovation or advancement in approach. The findings, while region-specific, do not significantly differ from prior research, diminishing their broader applicability. The focus on combined exposure events (THI, AQI, and BC) lacks sufficient clarity in establishing unique causal pathways, which limits the study's ability to provide actionable recommendations. The paper overemphasizes known relationships, such as the ‘U’-shaped THI curve, without exploring the underlying mechanisms in depth. Additionally, the definitions of combined events (T1, T2, T3) appear arbitrary and lack robust justification, raising concerns about the consistency and validity of results. The reliance on lag effects up to 3 days, though discussed extensively, does not sufficiently explore longer-term impacts, which could provide a more comprehensive understanding of respiratory mortality. The study's conclusions reiterate known risks without presenting significant new insights, while the suggestions for health early warning systems remain vague and impractical for implementation. Furthermore, the paper’s regional focus and reliance on single-area data limit its generalizability to other climate zones or socioeconomic contexts.

We look forward to receiving your revised manuscript.

Kind regards,

Venkatramanan S, Ph.D.

Academic Editor

PLOS ONE

Journal Requirements: When submitting your revision, we need you to address these additional requirements. 1. Please ensure that your manuscript meets PLOS ONE's style requirements, including those for file naming. The PLOS ONE style templates can be found at https://journals.plos.org/plosone/s/file?id=wjVg/PLOSOne_formatting_sample_main_body.pdf and https://journals.plos.org/plosone/s/file?id=ba62/PLOSOne_formatting_sample_title_authors_affiliations.pdf 2. We note that the grant information you provided in the ‘Funding Information’ and ‘Financial Disclosure’ sections do not match.  When you resubmit, please ensure that you provide the correct grant numbers for the awards you received for your study in the ‘Funding Information’ section. 3. Thank you for stating the following financial disclosure: "The 14th Five-Year Plan Distinctive Program of Public Health and Preventive Medicine in Higher Education Institutions of Xinjiang Uygur Autonomous Region" Please state what role the funders took in the study.  If the funders had no role, please state: ""The funders had no role in study design, data collection and analysis, decision to publish, or preparation of the manuscript."" If this statement is not correct you must amend it as needed. Please include this amended Role of Funder statement in your cover letter; we will change the online submission form on your behalf. 4. When completing the data availability statement of the submission form, you indicated that you will make your data available on acceptance. We strongly recommend all authors decide on a data sharing plan before acceptance, as the process can be lengthy and hold up publication timelines. Please note that, though access restrictions are acceptable now, your entire data will need to be made freely accessible if your manuscript is accepted for publication. This policy applies to all data except where public deposition would breach compliance with the protocol approved by your research ethics board. If you are unable to adhere to our open data policy, please kindly revise your statement to explain your reasoning and we will seek the editor's input on an exemption. Please be assured that, once you have provided your new statement, the assessment of your exemption will not hold up the peer review process. 5. PLOS requires an ORCID iD for the corresponding author in Editorial Manager on papers submitted after December 6th, 2016. Please ensure that you have an ORCID iD and that it is validated in Editorial Manager. To do this, go to ‘Update my Information’ (in the upper left-hand corner of the main menu), and click on the Fetch/Validate link next to the ORCID field. This will take you to the ORCID site and allow you to create a new iD or authenticate a pre-existing iD in Editorial Manager. 6. Please include captions for your Supporting Information files at the end of your manuscript, and update any in-text citations to match accordingly. Please see our Supporting Information guidelines for more information: http://journals.plos.org/plosone/s/supporting-information.

Reviewers' comments:

Reviewer's Responses to Questions

**Comments to the Author**

1. Is the manuscript technically sound, and do the data support the conclusions?

Reviewer #1: Yes

Reviewer #2: Yes

2. Has the statistical analysis been performed appropriately and rigorously? 

Reviewer #1: Yes

Reviewer #2: Yes

3. Have the authors made all data underlying the findings in their manuscript fully available?

Reviewer #1: Yes

Reviewer #2: Yes

4. Is the manuscript presented in an intelligible fashion and written in standard English?

Reviewer #1: No

Reviewer #2: Yes

5. Review Comments to the Author

Reviewer #1: This article presents a novel perspective on a relevant topic, closely aligning with current research hotspots. It offers a comprehensive, scientific, and detailed analysis, accompanied by an in-depth discussion and practical significance. Furthermore, it proposes feasible measures. Overall, this work meaningful. However, there remain several areas for improvement in the current version of the paper. More precisely.

I: Grammar problems and format problems

1. In the "Main Content" section, the formula for model construction, "Y = β0+β1X1+β2X2+ ... + βnXn+ε", requires improved spacing for clarity. It should be formatted as "Y = β0 + β1X1 + β2X2 + ... + βnXn + ε".

2. In the main section, the model builds, "Since the dependent and independent variables are continuous, we use the ordinary least squares regression model ols () function to build the model:" the colon should be a period to correctly end the sentence.

3. In section MAIN CONTENT, the spacing around mathematical formulas and symbols (such as "=" or "≥") shall be consistent. In some cases proper space around them.

4. Carefully modify the format of the reference.

II: the method of the rigorous inspection and suggestions

1. The study did not adequately detail the methods for processing missing data and outliers, which could potentially compromise the accuracy of the data analysis results. To enhance the reliability and scientific rigor of the study, it is recommended that the specific steps for data cleaning, as well as the statistical techniques employed to address missing data and outliers, be clearly outlined.

2. Although the article mentions the sensitivity analysis, it does not specify the specific content and results of the sensitivity analysis. It is recommended to add a detailed introduction of this part to increase the credibility of the conclusion.

III: the result part

The paper does not specify the construction process and rationality of the analytical model. The authors are advised to specify each step of model construction in the 'Materials and Methods' section and to discuss the applicability and limitations of the selected model.

IV: discussion part

1. In the discussion of practical applications, recommendations can be more comprehensive and specific, for example, how to combine the specific work of local health departments and how to communicate and cooperate with the public to improve the effectiveness of early warning measures.

2. Discuss one or more specific public health prevention strategies or interventions and how they can be integrated into the current health system.

Reviewer #2: The overall presentation of the manuscript is clear and the discussion is sufficient, but it is still recommended to modify the following to improve the quality of the manuscript:

The title of the article gives the illusion that it is a combination of three factors , but it is actually the interaction of THI and the other two that needs to be adjusted not to mislead the reader.

The abstract needs to be rewritten: T3A3, ER with full spelling and then abbreviations.

The specific location, total population, climate and demographic characteristics of the study are missing. What is the specific area?

Meteorological data and air quality data sources are downloaded or generated by yourself, to mark the reference and tell the generation process, is the data of several sites, etc., be more unclear.

In the statistical analysis section, FIG2 and TABLE2 should not appear and need to be modified.

wrong formula? g(X) = ols((y, df)+X)? Formula 2 dow has a coefficient y, right?

Joint meteorological events are categorical variables, binary classification? How to build a model to calculate RR should be explained in detail and attached with references.

The formula for calculating RR is 10*β?? Isn't it an IQR change?

And table 2 also not accurate, such as B1 does not contain B2? Is it an inclusive relationship?

Each analysis method should be marked with related references.

THI minimum 67 is not shown? The categorical values in Table 2 lack explanation and reason.

The resulting statement in Figure 2 is ER but the graph is RR? The figure also lacks confidence intervals and is incomplete.

The asterisk in Figure 3 is redundant.

Discussion to increase how these three factors might combine? Is it possible that weather interacts with each other.

minor: wrong text ? sexual?? hysteresis??

6. PLOS authors have the option to publish the peer review history of their article (what does this mean? ). If published, this will include your full peer review and any attached files.

**Do you want your identity to be public for this peer review?** For information about this choice, including consent withdrawal, please see our Privacy Policy .

Reviewer #1: No

Reviewer #2: **Yes: ** Hao Yanbin

---

## [Author Response · Author response to Decision Letter 1]

3 Jan 2025

List of Responses:

Dear Dr.academic editor and reviewers:

Thank you for your letter and for the reviewers’ comments concerning our manuscript

entitled “Exploring Health Effects under Specific Causes of Respiratory Mortality Based on

Different Definitions of Air Quality Index,Black carbon,Temperature and Humidity environment Combined Exposure in Southwest,China” (ID:PONE-D-24-49828 ). Those comments are all valuable and very helpful for revising and improving our paper, as well as the important guiding significance to our researches. We have studied comments carefully and have made correction which we hope meet with approval. Revised portion are marked in red in the paper.

The main corrections in the paper and the responds to the reviewer’s comments are as flowing:

Responds to the reviewer’s comments:

Academic editor

1.The major negative aspects of the paper include the lack of novelty in methodology, as the study relies heavily on existing models like DLNM and GAM without substantial innovation or advancement in approach.

2.The paper’s regional focus and reliance on single-area data limit its generalizability to other climate zones or socioeconomic contexts. The findings, while region-specific, do not significantly differ from prior research, diminishing their broader applicability. The paper overemphasizes known relationships, such as the ‘U’-shaped THI curve, without exploring the underlying mechanisms in depth.

3.The focus on combined exposure events (THI, AQI, and BC) lacks sufficient clarity in establishing unique causal pathways, which limits the study's ability to provide actionable recommendations.

4.Additionally, the definitions of combined events (T1, T2, T3) appear arbitrary and lack robust justification, raising concerns about the consistency and validity of results.

5.The reliance on lag effects up to 3 days, though discussed extensively, does not sufficiently explore longer-term impacts, which could provide a more comprehensive understanding of respiratory mortality.

6.The study's conclusions reiterate known risks without presenting significant new insights, while the suggestions for health early warning systems remain vague and impractical for implementation.

Response to academic editor

Thank you for your insightful comments. Your feedback is both constructive and practical, significantly contributing to the enhancement of the article's quality. We will address each of your comments and suggestions in our revisions. Our responses are as follows:

1.Thank you very much for your evaluation and suggestions. Prior to conducting our research, our team analyzed a substantial number of similar articles and consulted experts in relevant fields. In conjunction with our own research, we determined that the two models, DLNM and GAM, are the most suitable methods for our study. Currently, we are acquainted with emerging methodologies. For instance, the Bayesian Kernel Machine Regression Model (BKMR) has been developed in recent years for the analysis of pollutants. However, this model is not yet fully matured. Notably, it is primarily focused on correlation analysis between multiple pollutants, which does not align with our research methods. Additionally, the recently popular spatiotemporal models for pollutants are also inconsistent with our research data and methodologies. Consequently, the two models, DLNM and GAM, are the most appropriate methods for our research data and objectives. Notably, both methods are well-established and exhibit minimal shortcomings. We first employ the GAM to analyze nonlinear relationships in time series data, fitting potential patterns and trends using various smoothing functions. Subsequently, we utilize the DLNM to capture the dynamic changes in both time and dose effects, effectively addressing the complex relationships between environmental factors and health outcomes. Consequently, our data analysis is both effective and scientifically rigorous. Moreover, the majority of pertinent research in the field relies solely on one model for comprehensive analyses. In contrast, we integrate both models in our analysis, thereby leveraging their respective advantages. Thus, our analytical approach is both novel and practical. Based on our analysis, we will also elaborate on the benefits of integrating the two models in the methods section to enhance both the novelty and applicability of our approach. The supplementary content is as follows: Initially, we employ the GAM to analyze nonlinear relationships within time series data, effectively fitting potential patterns and trends using various smoothing functions. Subsequently, we utilize the DLNM to dynamically capture the temporal and dose-response effects, addressing the intricate relationships between environmental factors and health outcomes. This approach culminates in the establishment of an exposure-lag-response model. When used in conjunction, the two can fully leverage their complementary characteristics, thereby enhancing the novelty, depth, and breadth of the analysis.

2.Thanks very much to the expert for this advice.

①While this suggestion is valuable, it also represents a limitation of our study, specifically regarding regional applicability. The results of our research and the models developed for the early warning systems cannot be directly extrapolated to regions in different climate zones. This limitation arises from the fact that our study was conducted in an area characterized by consistently high temperatures and humidity throughout the year, conditions that may not be present in other climate zones. The primary aim of our research is to analyze the impact of combined exposure events to THI, AQI, and BC on respiratory mortality within the population. The methodologies, strategies, and findings of our study can serve as a reference for other regions, which is also a significant objective of our research. Additionally, we have incorporated a discussion on the limitations of our study, informed by expert opinions, as follows: To enhance and broaden the findings of this study, it is essential to incorporate additional research conducted in diverse climate zones. Additional relevant studies are necessary to enhance the reliability of these findings.

②Before conducting this study, we performed an extensive literature review and in-depth exploration of works in related fields. Unfortunately, we did not identify any relevant literature reports that utilized a combination of THI, AQI, and BC to investigate their interactions and establish an early warning system. Our study focuses on a specific area, and the results are derived from the distinct research, analysis, and discussion of three types of factors. However, based on the rigorous scientific data and analysis presented in this article, which aligns with the findings of most studies, the general outcome suggests a broader applicability. But we also mentioned in the article discussion section:Notably, the THI threshold results diverge from the conventional grading definitions. The average concentration and variation range of BC observed in this study were lower than those reported in previous studies. And explained the reason: This discrepancy may be attributed to China's vigorous efforts to control air pollution in recent years. However ,we have incorporated comparable research findings from other regions in alignment with your insights, as detailed below: Liang also found that, compared with high temperature conditions, hot and humid weather in Shanghai was significantly associated with a higher risk of outpatient visits and an increased risk of all-cause morbidity15. However, these findings are inconsistent with the research results of Fallah Ghalhari G, who found that the heat stress excitation threshold is inversely related to mortality29.This discrepancy may be attributed to the lower humidity levels in the Middle East, as well as variations in socioeconomic factors among the study samples. The primary focus of our article is to examine the impact of combined exposure events of THI, AQI, and BC on respiratory mortality outcomes. To our knowledge, there are limited studies addressing the combined effects of these three factors. Therefore, in this article, we present the results of all combined exposure events, explore potential influencing factors and causes, and discuss prospects for future research in the Discussion section.

③The impact of the THI on the population exhibits a 'U' shaped relationship, as THI is derived from a combination of daily average temperature and relative humidity. One notable characteristic of this relationship is the existence of an optimal interval. This finding aligns with the predominant results in scientific research and is consistent with reports on THI from other regions, thereby reinforcing the validity of our data and the scientific rigor of our analysis. Additionally, we have incorporated relevant supplements regarding the mechanism of action of THI based on your feedback, as detailed below: Although the biological mechanisms underlying thermohumidity events remain unclear, one possible explanation is that during low THI events, reduced humidity impairs human respiratory mucosal function and ciliary motility. This impairment facilitates the adherence of bacteria to the mucosa, thereby increasing the risk of infection30. An experiment demonstrates that mice exposed to low humidity exhibit impaired mucociliary clearance, diminished innate antiviral defenses, and compromised tissue repair, resulting in increased susceptibility to inflammasome caspase-mediated diseases31. During high THI events, elevated temperatures combined with high humidity can hinder the body's ability to dissipate heat. This may result in heat-related illnesses such as heat stroke and heat cramps, while also increasing the strain on the human respiratory system32,33

30.Liener K, Leiacker R, Lindemann J, et al. Nasal mucosal temperature after exposure to cold, dry air and hot, humid air. Acta Otolaryngol. 2003 Sep;123(7):851-6. doi: 10.1080/00016480310000601a

31.Kudo E, Song E, Yockey LJ, et al. Low ambient humidity impairs barrier function and innate resistance against influenza infection. Proc Natl Acad Sci U S A. 2019 May 28;116(22):10905-10910. doi: 10.1073/pnas.1902840116.

32.Fouillet A, Rey G, Laurent F,et al. Excess mortality related to the August 2003 heat wave in France. Int Arch Occup Environ Health. 2006 Oct;80(1):16-24. doi: 10.1007/s00420-006-0089-4.

33. Kenny GP, Wilson TE, Flouris AD, et al. Heat exhaustion. Handb Clin Neurol. 2018;157:505-529. doi: 10.1016/B978-0-444-64074-1.00031-8.

3.Thank you for your valuable views, questions, and comments. Firstly, based on your feedback, our team re-evaluated the entire experimental design and process. This included reorganizing the data, selecting the optimal method according to the characteristics and type of data, and conducting scientific and rigorous modeling calculations, which yielded consistent results. Consequently, we affirm that our research is accurate, and the causal relationships are overall clear and unambiguous. Secondly, we conducted an extensive review of the literature and found that the results of our research methods align with those of other similar studies. Lastly, we meticulously reviewed the discussion section and identified certain logical inconsistencies that could lead to confusion regarding cause and effect. We sincerely appreciate your insights. In response to your and other reviewers' comments, we have thoroughly supplemented each modeling and calculation step in our research design to ensure that the causal pathways are clearly articulated. The supplementary content is as follows: We meticulously cleaned, screened, and organized the data in alignment with the objectives of the research. Initially, specific time periods and diseases were identified, and data were excluded for individuals who were not present in the area during their lifetime, as well as for those who resided in the area for less than five years prior to their death. We subsequently removed any missing values and cross-verified values that deviated by more than three standard deviations from the long-term mean. Based on your feedback, we have revised certain statements in the discussion section that contained confusing expressions and reversed logic. The modified sections are highlighted in red within the article.

4.We appreciate your valuable feedback. This oversight in our writing has been addressed. In response to your suggestions, we have provided a more detailed explanation of the definitions and underlying reasons in the statistical methods section, as follows: The definition of exposure events is informed by the RCS exposure response curve, in conjunction with the 'GB3095-2012' standard established by the World Meteorological Organization (WMO), the National Meteorological Administration, and the China Meteorological Administration for heat wave definitions, as well as previous research. We categorize the Temperature-Humidity Index (THI) as follows: T1 (THI < 63), T2 (63 ≤ THI < 75), and T3 (THI ≥ 75). Furthermore, by integrating the BC and AQI percentiles from this study and referencing the Technical Specifications for Ambient Air Quality Assessment (HJ6632013), the Guidelines for the Preparation of Urban Heavy Air Pollution Emergency Plans, and the Sichuan Province Heavy Pollution Weather Emergency Plan (2018 Revision) - Quantitative Indicators, we redefine the BC and AQI categories as B1 (P≥50), B2 (P≥75), B3 (P≥90), A1 (P≥50), A2 (P≥75), and A3 (P≥90). Notably, A1 encompasses A2 and A3, while B1 includes B2 and B3. This study utilizes the RCS intersection value as a node, treating the combined meteorological event variable as a continuous variable, thereby fixing it within the ranges above or below different percentiles. This approach aims to investigate the relationship between THI and extreme AQI and BC, as well as to explore the impact of exposure to combined events with varying threshold ranges on respiratory mortality, ultimately refining the thresholds for early warning models.

5. Thank you very much for your professional review. Our team also evaluated the necessity of long-term exploration at the outset of the research. Ultimately, however, we decided against pursuing long-term exploration in this article based on the study's objectives. Firstly, the meaningful results derived from long-term exploration span an extensive timeframe, which does not align with our research goals or the requirement for establishing an early warning system. Additionally, our investigation into short-term effects has yielded significant and practical results, accompanied by a rich discussion that sufficiently supports the design and objectives of our study. Finally, in the context of establishing an early warning system, short-term effects are most relevant. Long-term effects necessitate prolonged follow-up of the exposed population, which does not correspond with the aims of our study. Therefore, we opted not to explore long-term effects in this paper. However, in response to your suggestions, we have revised the title and limitations to more effectively present the short-term effects. The modifications are as follows: Exploring Temperature and Humidity Environment Combined with Air Quality Index,Black carbon,the short-term Effect of Combined Exposure on Respiratory Disease Mortality in Southwest China.

Additionally, this study primarily concentrated on short-term effects and did not fully investigate the long-term outcomes related to combined exposure events and respiratory deaths. This limitation may affect the comprehensiveness of the findings. To enhance and broaden the findings of this study, it is essential to incorporate additional research conducted in diverse climate zones, additional relevant studies are necessary to enhance the reliability of these findings.

6.We appreciate your insightful feedback. Our findings regarding individual exposure events align with previous research, underscoring the significance of our study. Given the limited literature on combined exposure events involving THI, AQI, and BC, much of the discourse presented in this article represents novel research contributions. We a

---

## [Decision Letter · Decision Letter 1]

5 Feb 2025

Exploring Health Effects under Specific Causes of Respiratory Mortality Based on Different Definitions of Air Quality Index,Black carbon,Temperature and Humidity environment Combined Exposure in Southwest,China

PONE-D-24-49828R1

Dear Dr. Xie,

We’re pleased to inform you that your manuscript has been judged scientifically suitable for publication and will be formally accepted for publication once it meets all outstanding technical requirements.

Kind regards,

Venkatramanan S, Ph.D.

Academic Editor

PLOS ONE

Additional Editor Comments (optional):

Reviewers' comments:

Reviewer's Responses to Questions

**Comments to the Author**

1. If the authors have adequately addressed your comments raised in a previous round of review and you feel that this manuscript is now acceptable for publication, you may indicate that here to bypass the “Comments to the Author” section, enter your conflict of interest statement in the “Confidential to Editor” section, and submit your "Accept" recommendation.

Reviewer #1: All comments have been addressed

Reviewer #2: All comments have been addressed

2. Is the manuscript technically sound, and do the data support the conclusions?

Reviewer #1: Yes

Reviewer #2: Yes

3. Has the statistical analysis been performed appropriately and rigorously? 

Reviewer #1: Yes

Reviewer #2: Yes

4. Have the authors made all data underlying the findings in their manuscript fully available?

Reviewer #1: Yes

Reviewer #2: No

5. Is the manuscript presented in an intelligible fashion and written in standard English?

Reviewer #1: Yes

Reviewer #2: Yes

6. Review Comments to the Author

Reviewer #1: The author has answered my concerns very well. Thanks the authors have addressed all reversion. I agree to publish it in its present form.

Reviewer #2: I now approve the manuscript for publication because the authors have made every effort to clearly present their research and results.

7. PLOS authors have the option to publish the peer review history of their article (what does this mean? ). If published, this will include your full peer review and any attached files.

**Do you want your identity to be public for this peer review?** For information about this choice, including consent withdrawal, please see our Privacy Policy .

Reviewer #1: No

Reviewer #2: **Yes: ** Yanbin Hao

---

## [Editor Report · Acceptance letter]

PONE-D-24-49828R1

PLOS ONE

Dear Dr. Xie,

I'm pleased to inform you that your manuscript has been deemed suitable for publication in PLOS ONE. Congratulations! Your manuscript is now being handed over to our production team.

Kind regards,

on behalf of

Dr. Venkatramanan S

Academic Editor

PLOS ONE